# Application of glutamate weighted CEST in brain imaging of nicotine dependent participants *in vivo* at 7T

**Paul S. Jacobs**[1]*, **Joelle Jee**[2], **Liu Fang**[3], **Emily Devlin**[4], **Claudia Iannelli**[5], **Deepa Thakuri**[4], **James Loughead**[4], **Cynthia Neill Epperson**[5], **Neil Wilson**[1], **David Roalf**[2], **Ravinder Reddy**[1], **Ravi Prakash Reddy Nanga**[1]*

1 Center for Advanced Metabolic Imaging in Precision Medicine, Department of Radiology, University of Pennsylvania, Philadelphia, PA, United States of America, 2 Department of Psychiatry, University of Pennsylvania, Philadelphia, PA, United States of America, 3 Penn Statistics in Imaging and Visualization Center, Department of Biostatistics, Epidemiology, and Informatics, University of Pennsylvania, Philadelphia, PA, United States of America, 4 Perelman School of Medicine, University of Pennsylvania, Philadelphia, PA, United States of America, 5 Department of Psychiatry, University of Colorado School of Medicine, Anschutz Medical Campus, Aurora, CO, United States of America

* pauljaco@seas.upenn.edu (PSJ); nravi@pennmedicine.upenn.edu (RPRN)

**Data Availability Statement:** All relevant data are within the paper and its Supporting Information files.

## Abstract

### Introduction

With nicotine dependence being a significant healthcare issue worldwide there is a growing interest in developing novel therapies and diagnostic aids to assist in treating nicotine addiction. Glutamate (Glu) plays an important role in cognitive function regulation in a wide range of conditions including traumatic brain injury, aging, and addiction. Chemical exchange saturation transfer (CEST) imaging *via* ultra-high field MRI can image the exchange of certain saturated labile protons with the surrounding bulk water pool, making the technique a novel tool to investigate glutamate in the context of addiction. The aim of this work was to apply glutamate weighted CEST (GluCEST) imaging to study the dorsal anterior cingulate cortex (dACC) in a small population of smokers and non-smokers to determine its effectiveness as a biomarker of nicotine use.

### Methods

2D GluCEST images were acquired on 20 healthy participants: 10 smokers (ages 29–50) and 10 non-smokers (ages 25–69), using a 7T MRI system. $T_1$-weighted images were used to segment the GluCEST images into white and gray matter tissue and further into seven gray matter regions. Wilcoxon rank-sum tests were performed, comparing mean GluCEST contrast between smokers and non-smokers across brain regions.

### Results

GluCEST levels were similar between smokers and non-smokers; however, there was a moderate negative age dependence ($R^2 = 0.531$) in smokers within the cingulate gyrus.

**Funding:** This work was supported by a pilot grant from the Thomas B. and Jeanette E. Laws McCabe Fund to RPRN, the National Institutes of Biomedical Imaging and Bioengineering of the National Institutes of Health under Award Number P41EB029460 to RR, the National Institute of Mental Health of the National Institutes of Health under Award Numbers MH120174 to DR and MH119185 to DR, and the National Institute of Drug Abuse of the National Institutes of Health under Award Number R01 DA037289 to CNE and R01 DA018359 to CNE. The funders had no role in study design, data collection and analysis, decision to publish, or preparation of the manuscript. There was no additional external funding received for this study.

**Competing interests:** I have read the journal's policy and the authors of this manuscript have the following competing interests: Dr. Epperson discloses that she has received research grant support from Sage Therapeutics and HealthRhythms. She is a consultant for Sage Therapeutics, BabyScripts, and Asarina Pharma. This does not alter our adherence to PLOS ONE policies on sharing data and materials.

## Conclusion

Feasibility of GluCEST imaging was demonstrated for *in vivo* investigation of smokers and non-smokers to assess glutamate contrast differences as a potential biomarker with a moderate negative age correlation in the cingulate gyrus suggesting reward network involvement.

## Introduction

As an excitatory neurotransmitter, glutamate (Glu) plays an important role in regulating various motor and cognitive functions and is also implicated in brain injury, substance abuse, and aging [1, 2]. Glu receptors have been a target in nicotine addiction research with a recent focus on metabotropic Glu receptor-5 (mGluR5) [3–7]. Nicotine dependence remains a major worldwide public healthcare issue. Since it is one of the most preventable causes of chronic illness and premature death, there is a growing interest in developing novel therapies as well as diagnostic aids to treat and monitor nicotine addiction. Several groups have investigated the relationship between nicotine dependence and Glu concentrations. A study by O'Neill et al. [8] investigated the relationship between thalamic Glu and smoking *via* proton magnetic resonance spectroscopy ($^1$H MRS) at 1.5T. The authors observed no statistical difference in thalamic Glu levels between smokers and non-smokers. However, this study was limited by the small sample size as well as the large voxel sizes used during MRSI acquisition which prevented the analysis of separate brain regions. Gallinat et al. [9] used $^1$H MRS on the left hippocampus and anterior cingulate cortex (ACC) in chronic smokers, former smokers, and non-smokers at 3T. Results indicated no significant correlations between Glu concentrations and the age at which the smoker began smoking, daily nicotine exposure, or overall lifetime nicotine exposure. This study was limited by the small sample size and lack of spatial specificity, in the form of two relatively large voxels. More recently, however, Durazzo et al. [10] performed $^1$H MRS on the perigenual anterior cingulate cortex (ACC) and the right dorsolateral prefrontal cortex (DLPFC) at 4T and spectroscopic imaging at 1.5T. The authors showed a decrease in DLPFC Glu levels and a greater age-related decrease in ACC and DLPFC Glu levels in smokers compared to non-smokers, however were not able to assess the effect of sex on the results due to limited female subjects. All of these studies were performed using almost exclusively spectroscopic techniques and therefore were not able to simultaneously measure Glu levels across multiple regions of the brain. Furthermore, the low field strength commonly used in these previous studies is a major confounding factor as it does not allow for adequate chemical shifts to properly resolve and quantify the spectroscopic Glu peak. It has recently been demonstrated that Glu can be measured with higher sensitivity and at higher spatial resolution with glutamate-weighted chemical exchange saturation transfer (GluCEST) imaging at 7T [11]. This magnetic resonance imaging (MRI) technique can image specific labile protons by saturating the chemical species of interest (in this case the amine protons of Glu), with a spectrally selective saturation pulse and allowing the saturated protons to transfer to the bulk water pool over time. This process of saturation transfer is continuously repeated which leads to a buildup of saturation in the bulk water magnetization, which can then be measured [11–15]. In the present work, GluCEST imaging was applied to image regions of the reward network in the dorsal anterior cingulate cortex (dACC) to determine its effectiveness as an imaging biomarker of nicotine addiction in a population of cigarette smokers and non-smokers.

## Methods

### Human participants

Imaging data were acquired on a total of 20 healthy volunteers, under an approved University of Pennsylvania Institutional Review Board protocol in which written consent was obtained. These participants were recruited during the period between December 11[th], 2018, and October 15[th], 2019 and were divided between non-smokers (n = 10; 5 Females, 5 Males; age = 44.2 ±14.87 years) and smokers (n = 10; 5 Females, 5 Males; age = 39.7±8.37 years; 5 ≤ cigarettes/ day). Data from one participant in the smoker participants group was not usable due to severe motion artifacts and was excluded from analysis.

Key inclusion criteria used for subject recruitment included: participants between the ages of 20–70 years old with 1 female and 1 male in each age decade (smokers between the ages of 20–50 years old), generally good health as per self-report, negative urine drug screen with the exception of marijuana, self-reported non-smoking individuals having an expired carbon monoxide (CO) level of below 8ppm, and in the case of those who smoke, individuals smoking five or more cigarettes a day and having an expired CO level of at least 8ppm.

Key exclusion criteria included: the presence or history of severe or unstable physical or neurological disorders as per self-reported medical history, current use of psychotropic medication, substance abuse within the previous two years, current untreated or undertreated axis I disorder, as per structured clinical interview using the Structured Clinical Interview for DSM Disorders-IV (SCID-IV) and the discretion of the principal investigator or clinical designee, lifetime history of psychotic disorder as per SCID-IV interview, pregnancy or planning to become pregnant, claustrophobia or metal in the body, and current use of hormone contraceptives.

### MR imaging data acquisition

All images were acquired on a 7T system (Magnetom Terra, Siemens Healthcare, Erlangen, Germany) using a single volume transmit 32-channel receive phased-array head coil (Nova Medical, Wilmington, MA, USA). Across all participants repeatable oblique slice placement was performed by aligning the imaging slice to be in-plane with the main body of the dACC just superior to the corpus callosum. Hand placement of this imaging slice was performed to account for any head tilt between subjects. A 2D GluCEST acquisition was performed with the following parameters: slice thickness = 5mm, in-plane resolution = 1x1mm$^2$, matrix size = 240x168, GRE readout TR/TE = 3.3/1.48ms, bandwidth = 690Hz/Px, duty-cycle = 99, flip-angle = 6˚, 2 averages, shot TR = 6000ms, shots-per-slice = 1, and a $B_{1.RMS}$ = 3.06μT with an 800ms long saturation pulse (100ms pulse train). CEST images were acquired by varying the saturation offset frequency for ±1.8ppm to ±4.2ppm (relative to the water resonance set to 0ppm) with a step-size of 0.3ppm. To compute $B_0$ maps for correction of $B_0$ inhomogeneity, water saturation shifting referencing (WASSR) images [16] were collected from 0 to ±1.5ppm (step-size of 0.15ppm) with a $B_{1.RMS}$ = 0.29μT and a 200ms pulse duration. A relative $B_1$ map was generated from three images using square saturation pulses with flip-angles of 20˚, 40˚, and 80˚, as described in Volz et al. [17]. The total acquisition time for CEST images, $B_0$, and $B_1$ field maps was approximately 12 minutes. GluCEST contrast maps in the imaging slice were generated after $B_0$ and $B_1$ inhomogeneity corrections in MATLAB (The Mathworks Inc., Natick, MA, USA) were applied, as shown in Fig 1A. A 3D Magnetization Prepared Rapid Gradient Echo (MP2RAGE) MRI was acquired with the same spatial parameters used in the CEST sequence to generate a $T_1$ map for use in tissue region of interest (ROI) segmentation. Automatic segmentation was performed using the FSL FAST image segmentation tool [18] to yield

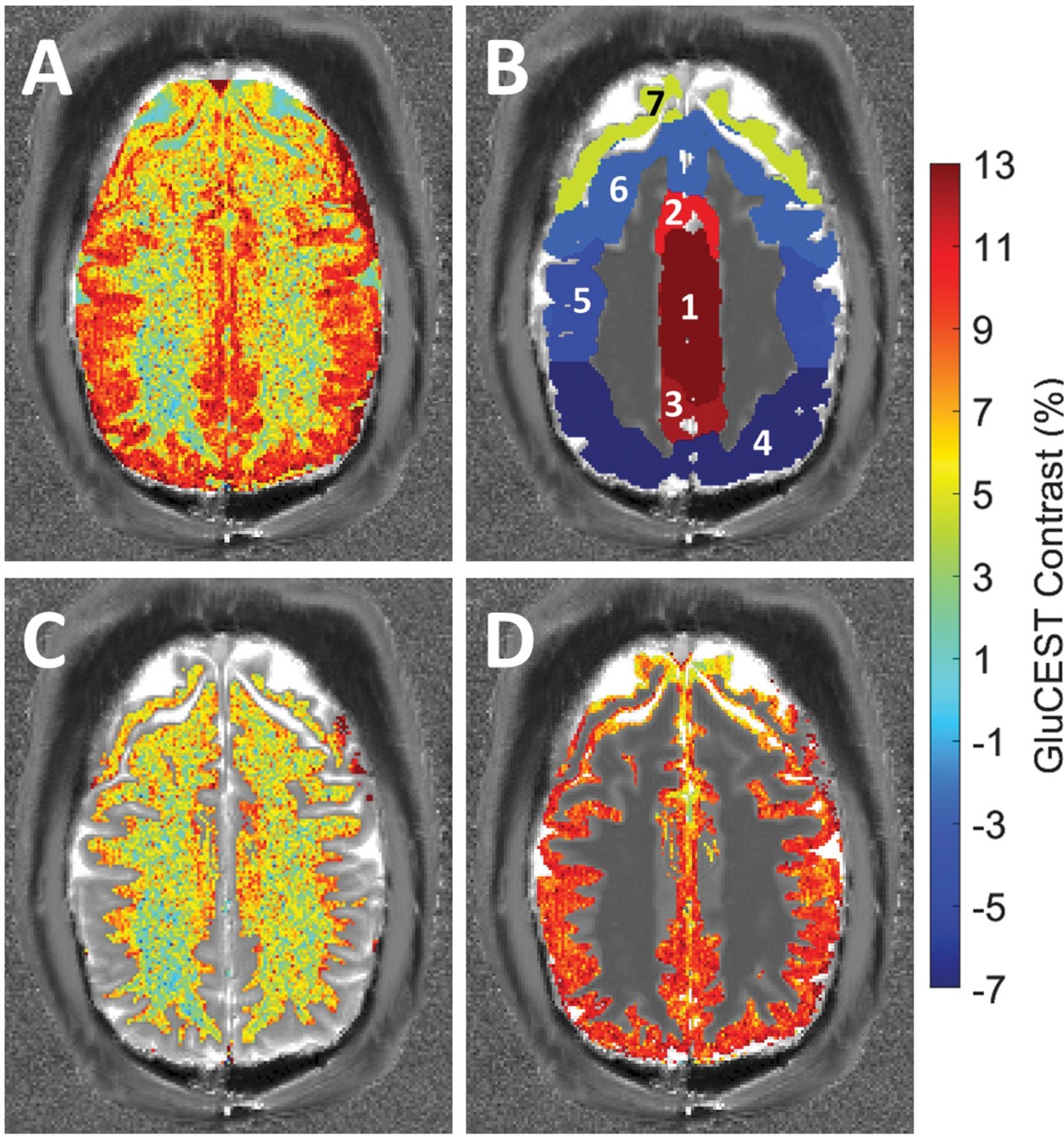

**Fig 1. Representative GluCEST contrast image and segmentation regions.** A) An example of a GluCEST contrast map of the dorsal anterior cingulate cortex and B) the same GluCEST slice with numbered ROIs overlaid corresponding to: (1) cingulate gyrus, (2) juxtapositional lobule cortex, (3) paracingulate gyrus, (4) frontal pole, (5) middle frontal gyrus, (6) precentral gyrus, (7) postcentral gyrus. C) White matter and D) gray matter GluCEST segmentations can also be seen.

separate gray and white matter regions (Fig 1C and 1D respectively) along with seven separate gray matter tissue segmentations (Fig 1B) for analysis (cingulate gyrus, juxtapositional lobule cortex, paracingulate gyrus, frontal pole, middle frontal gyrus, precentral gyrus, postcentral gyrus). Correlation between normalized gray and white matter volumes (normalized to overall

brain volume) [19] to the corresponding average GluCEST contrast was also performed to assess any potential relationships with the additional covariate.

## Statistical analysis

Wilcoxon rank-sum tests were performed to compare mean GluCEST contrast between smokers and non-smokers across the global slice, subcortical segmentations, and the seven cortical regions with p-values reported. A linear regression model was used to test the significance between potentially correlated variables such as average GluCEST contrast and age. All statistical analyses were conducted using R statistical environment (version 4.1.1).

## Results

The full acquisition of GluCEST images for both smoking and non-smoking participants can be seen presented in Fig 2, in which no immediate qualitative contrast differences can be seen. Segmented GluCEST contrast values are plotted across all participants as histogram comparisons in Fig 3. These all show similar GluCEST distributions across gray and white matter tissue types as well as separate cortical segmentation regions. No statistically significant difference in mean GluCEST contrast values was observed between smoker and non-smokers for any region. This is quantified in Table 1 where average and standard deviation GluCEST contrast values along with corresponding statistical p-values are presented as: whole brain (p = 0.44), gray matter (p = 0.90), white matter (p = 0.78), cingulate gyrus (p = 0.96), juxtapositional lobule cortex (p = 0.71), paracingulate gyrus (p = 0.77), frontal pole (p = 0.59), middle frontal gyrus (p = 0.54), precentral gyrus (p = 0.56), and postcentral gyrus (p = 0.90).

Average GluCEST contrast values were also taken into consideration across subject ages, as shown in Fig 4, with linear regression lines overlaid. GluCEST contrast in the non-smokers showed weak to very weak correlations relative to age across the brain regions observed: (gray matter ($R^2$ = 0.046), white matter ($R^2$ = 0.020), cingulate gyrus ($R^2$ = 0.001), juxtapositional lobule cortex ($R^2$ = 0.353), paracingulate gyrus ($R^2$ = 0.011), frontal pole ($R^2$ = 0.057), middle frontal gyrus ($R^2$ = 0.001), precentral gyrus ($R^2$ = 0.001), and postcentral gyrus ($R^2$ = 0.062)), GluCEST contrast in smokers showed moderate to very weak correlations relative to age across the brain regions observed: (gray matter ($R^2$ = 0.186), white matter ($R^2$ = 0.123), cingulate gyrus ($R^2$ = 0.531), juxtapositional lobule cortex ($R^2$ = 0.0112), paracingulate gyrus ($R^2$ = 0.338), frontal pole ($R^2$ = 0.097), middle frontal gyrus ($R^2$ = 0.022), precentral gyrus ($R^2$ = 0.005), and postcentral gyrus ($R^2$ = 0.009)). A linear regression model was performed between age and the region with the highest $R^2$ value, the smoker cingulate gyrus, and yielded a p-value = 0.04, indicating statistical significance. Lastly, correlations between gray and white matter volumes to average GluCEST contrast values can be seen in Fig 5, with linear regression lines overlaid. GluCEST contrast in smokers showed very weak correlations relative to gray matter ($R^2$ = 0.02) and white matter ($R^2$ = 0.02) volumes. Non-smokers also showed very weak to weak correlations relative to gray matter ($R^2$ = 0.34) and white matter ($R^2$ = 0.08) volumes.

## Discussion

The aim of this work was to demonstrate the feasibility of the GluCEST technique as a potential imaging biomarker for mapping this portion of the brain reward network in nicotine addiction between cigarette smokers and non-smokers. In the work presented here, while there was one statically significant correlation between age and GluCEST in the cingulate gyrus of the smokers, no statistically significant difference was observed in any of the measured cortical or subcortical regions between smokers and non-smokers in this limited pilot study.

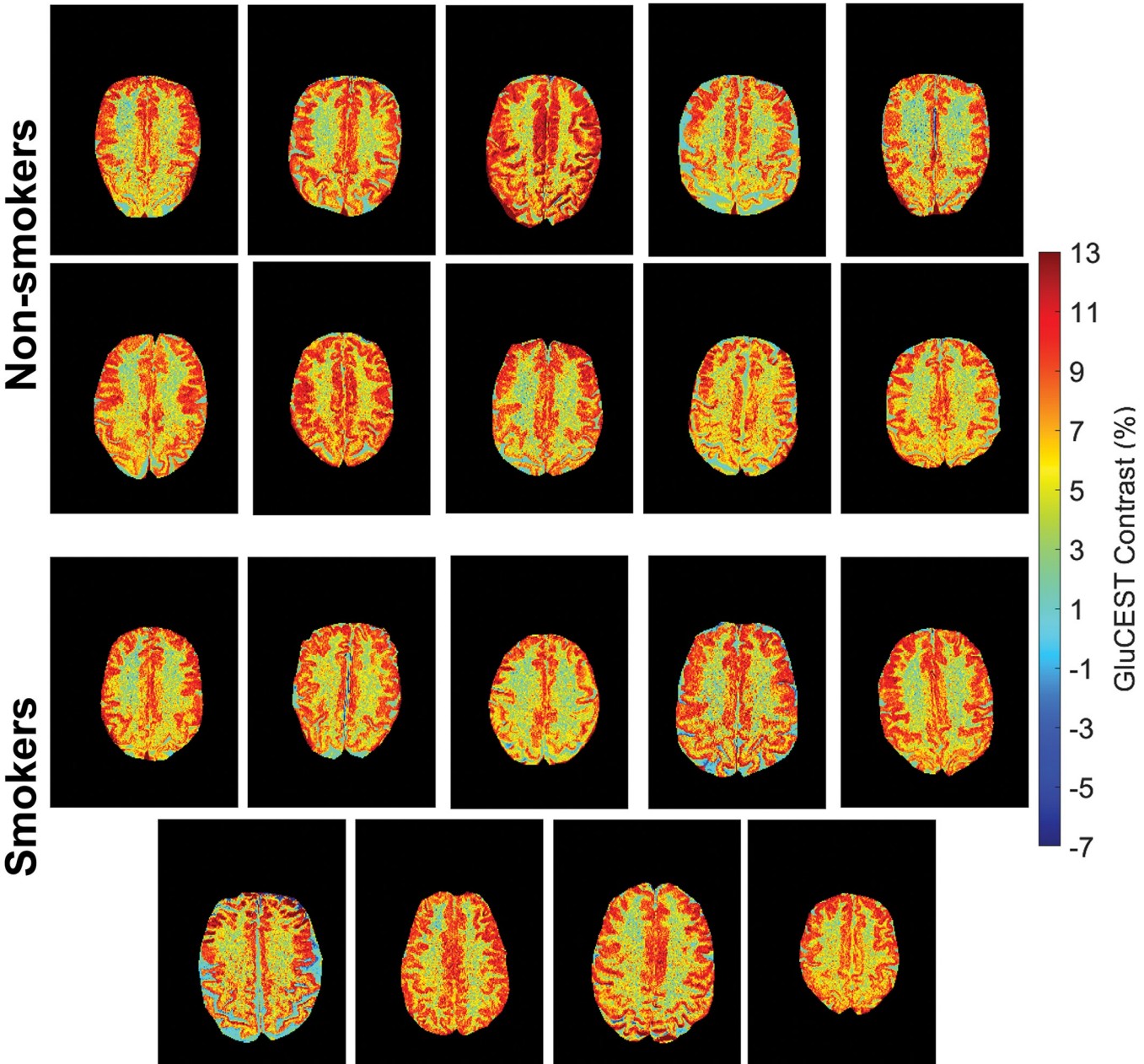

**Fig 2. Complete GluCEST contrast image set.** All 2D GluCEST images acquired on the dorsal anterior cingulate cortex (dACC) for the 10 non-smokers and 9 smokers.

When evaluating GluCEST age dependence, weak correlations were observed between smokers and non-smokers. This could be due to a limited sample of one male and one female per decade between 20 and 70 years. One region that did show a moderate negative age dependence with a statistically significant correlation ($p = 0.04$), was in the cingulate gyrus in smokers ($R^2 = 0.531$). One possible explanation could be that early and acute nicotine exposure leads to a transient increase of Glu through excitatory nicotinic acetylcholine (nACh) receptors located on presynaptic glutamatergic terminals [20]. This, in the context of chronic nicotine exposure, would lead to excitotoxic cell death which would lead to overall lower Glu levels

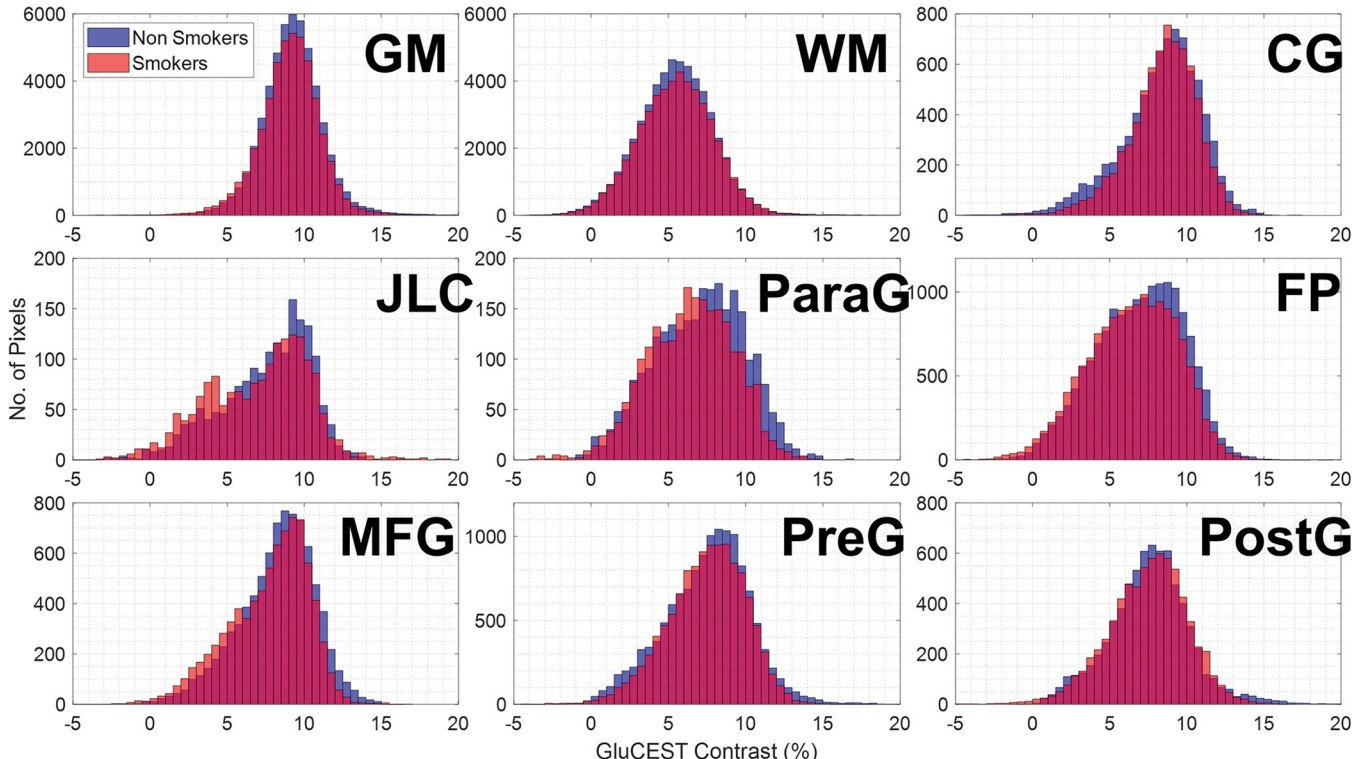

**Fig 3. Quantitative GluCEST contrast comparison.** Histogram distribution comparisons of GluCEST contrast values between smokers and non-smokers for gray matter (GM), white matter (WM), and the seven cortical segmentations including: cingulate gyrus (CG), juxtapositional lobule cortex (JLC), paracingulate gyrus (ParaG), frontal pole (FP), middle frontal gyrus (MFG), precentral gyrus (PreG), and postcentral gyrus (PostG).

**Table 1. GluCEST statistical results.**

| Brain Region | Smoker Status | GluCEST Contrast (%) | p-value |
|---|---|---|---|
| Whole Brain | NS | 7.07 (7.18) | 0.44 |
| | S | 7.06 (12.03) | |
| GM | NS | 9.19 (1.50) | 0.90 |
| | S | 9.05 (3.61) | |
| WM | NS | 8.60 (4.24) | 0.78 |
| | S | 7.88 (4.82) | |
| CG | NS | 8.95 (0.81) | 0.96 |
| | S | 8.99 (0.67) | |
| JLC | NS | 9.04 (0.43) | 0.71 |
| | S | 8.98 (0.80) | |
| ParaG | NS | 9.21 (0.56) | 0.77 |
| | S | 9.12 (0.37) | |
| FP | NS | 9.19 (0.39) | 0.59 |
| | S | 9.10 (0.25) | |
| MFG | NS | 9.62 (0.77) | 0.54 |
| | S | 9.31 (0.43) | |
| PreG | NS | 9.02 (0.69) | 0.56 |
| | S | 8.84 (0.38) | |

*(Continued)*

**Table 1.** (Continued)

| Brain Region | Smoker Status | GluCEST Contrast (%) | p-value |
|---|---|---|---|
| PostG | NS | 8.33 (1.12) | 0.9 |
| | S | 8.67 (0.54) | |

Average and standard deviation GluCEST contrast values for non-smokers (NS) and smokers (S) across the whole brain, gray matter (GM), white matter (WM), cingulate gyrus (CG), juxtapositional lobule cortex (JLC), paracingulate gyrus (ParaG), frontal pole (FP), middle frontal gyrus (MFG), precentral gyrus (PreG), and postcentral gyrus (PostG). Statistical p-values are also presented for the comparison between groups.

through long-term cigarette usage. The cingulate gyrus may be primarily affected by this due to it being associated with reward anticipation [21]. In the context of this work, the wider literature about regional Glu difference in smokers compared to non-smokers is conflicted, with some studies showing a difference [10, 22] and others showing no difference [8, 9, 23]. These previous studies were all performed at 4T field strength and below and all used different MRS techniques, which do not allow for simultaneous measurement of Glu across multiple brain regions, unlike GluCEST imaging. These spectroscopic studies performed at lower field strengths also resulted in a lack of adequate chemical shift dispersion to properly resolve the

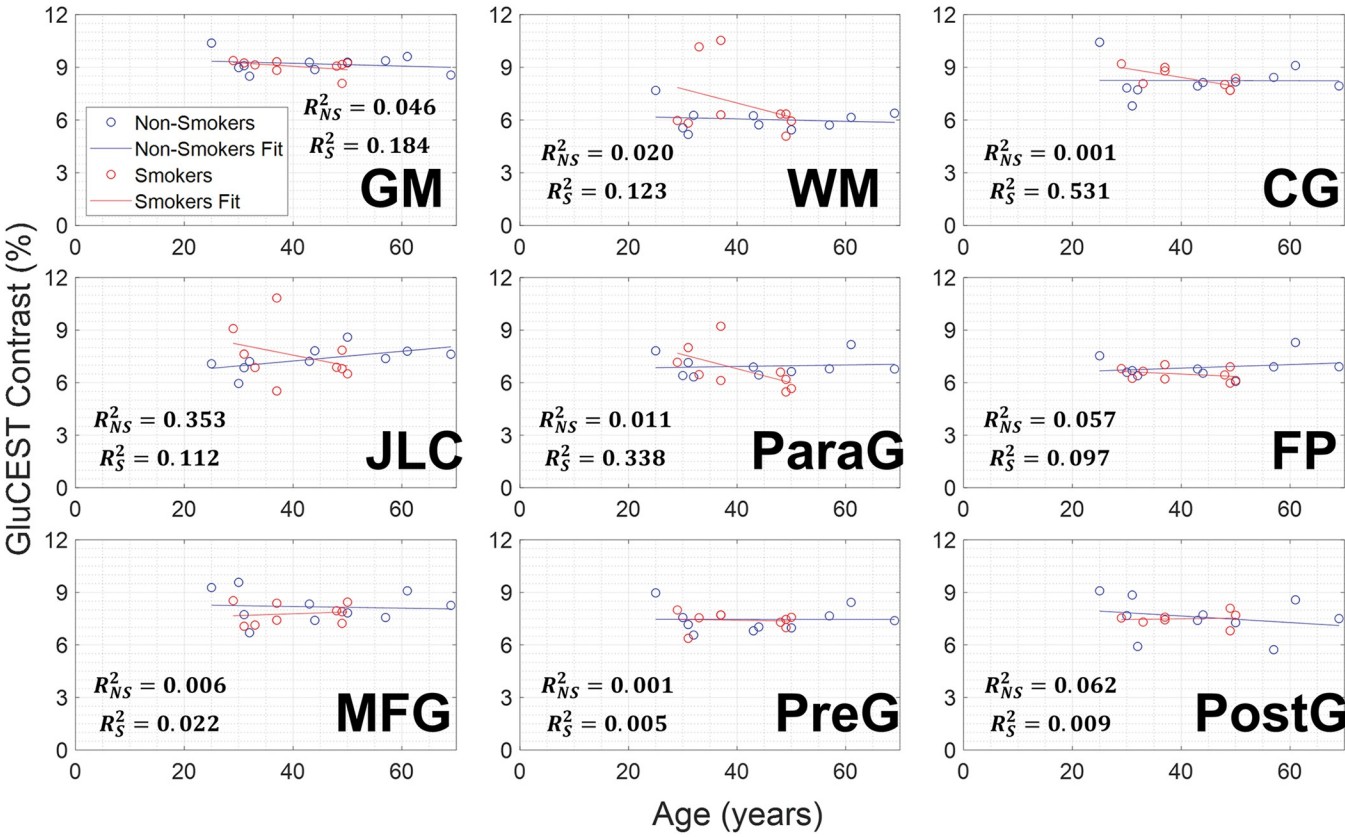

**Fig 4. Correlation between GluCEST contrast and subject age.** Plots showing the correlation between subject age and average GluCEST contrast value between smokers and non-smokers for gray matter (GM), white matter (WM), and the seven cortical segmentations including: cingulate gyrus (CG), juxtapositional lobule cortex (JLC), paracingulate gyrus (ParaG), frontal pole (FP), middle frontal gyrus (MFG), precentral gyrus (PreG), and postcentral gyrus (PostG). Linear regression fit lines are overlaid to show general trend comparisons.

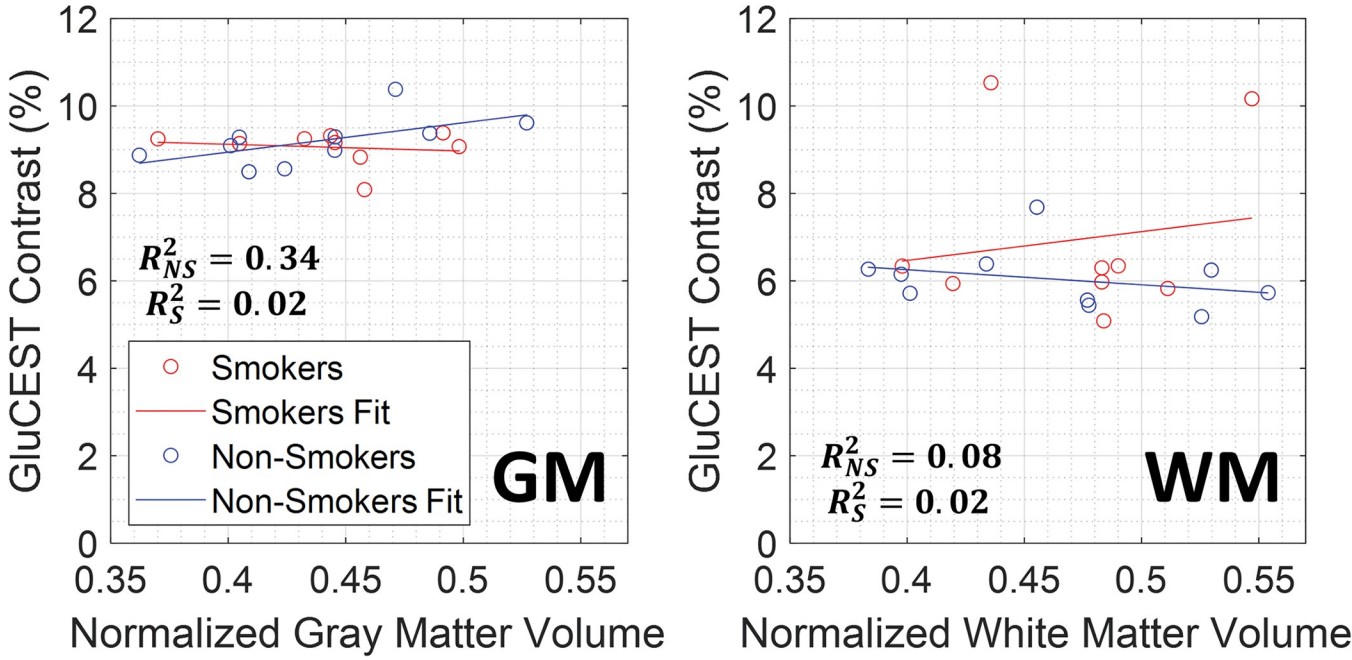

**Fig 5. Correlation between GluCEST contrast and subject brain volumes.** Plots showing the correlation between subject normalized gray and white matter volumes and average GluCEST contrast value between smokers and non-smokers. Linear regression fit lines are overlaid to show general trend comparisons.

Glu peak from the glutamine peak. Using GluCEST imaging at 7T, as was done in this study, was able to better address the technical limitations mentioned above.

There were several limitations within the work presented here. As this was a small initial pilot study, a small sample size of only one subject per sex in each major age decade (10 non-smokers, 9 smokers total) was present and decreased the ability to assess age-associated differences and reduced overall statistical power. With all smokers in the study smoking a minimum of 5 cigarettes/day, the amount of nicotine exposure may not have been elevated enough to produce a substantial difference in Glu compared to non-smoking individuals. In addition, image segmentation methods also reduced the number of brain regions that could be reliably assessed within the GluCEST map due to the underrepresentation of pixels in those regions caused in part by differences in brain size between participants. Imaging artifacts such as $B_1$ inhomogeneities (due to the shortened RF wavelengths used at 7T), $B_0$ inhomogeneities, or image partial voluming could have confounded GluCEST contrast quality and quantification. Lastly, single slice GluCEST also precluded us from investigating other major networks across the brain.

In the future, expanding the number of smoking and non-smoking participants to increase the power of the statistical testing would be beneficial in eliciting any potential subtle experimental effect sizes not seen in the current work. Additionally, single slice (2D) GluCEST is limited in the amount of data that can be captured in a single scan. With the recent introduction of volumetric (3D) GluCEST [11], partial voluming within the areas of interest can be reduced and a nominal brain slab volume of approximately 20mm can be reliably investigated. An added benefit of utilizing 3D GluCEST imaging would also include the capability to map the entire dACC/reward network rather than a limited single slice.

## Conclusion

In conclusion, the feasibility of GluCEST imaging was demonstrated *in vivo* for brain imaging of smokers and non-smokers to assess contrast differences as a potential biomarker. While one

subregion, the smoker cingulate gyrus, did show a statistically significant correlation between average GluCEST contrast and age the general comparison between smokers and non-smokers did not reach statistical significance.

## Supporting information

**S1 Data.**
(MAT)

**S2 Data.**
(XLSX)

## Author Contributions

**Conceptualization:** Claudia Iannelli, Cynthia Neill Epperson, Ravinder Reddy, Ravi Prakash Reddy Nanga.

**Data curation:** Paul S. Jacobs, Neil Wilson, Ravi Prakash Reddy Nanga.

**Formal analysis:** Paul S. Jacobs, Joelle Jee, Liu Fang.

**Funding acquisition:** Claudia Iannelli, Cynthia Neill Epperson, David Roalf, Ravinder Reddy, Ravi Prakash Reddy Nanga.

**Investigation:** Deepa Thakuri, Neil Wilson, David Roalf, Ravi Prakash Reddy Nanga.

**Methodology:** James Loughead, Cynthia Neill Epperson, Neil Wilson, Ravinder Reddy.

**Project administration:** Emily Devlin, Deepa Thakuri.

**Resources:** Neil Wilson, David Roalf.

**Software:** Liu Fang, Neil Wilson.

**Supervision:** Ravinder Reddy.

**Visualization:** Paul S. Jacobs.

**Writing – original draft:** Paul S. Jacobs, Ravi Prakash Reddy Nanga.

**Writing – review & editing:** Claudia Iannelli, James Loughead, Cynthia Neill Epperson, David Roalf, Ravinder Reddy, Ravi Prakash Reddy Nanga.

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
