## [Decision Letter · Decision Letter 0]

29 Nov 2023

PONE-D-23-22757Application of glutamate weighted CEST in brain imaging of nicotine dependent participants *in vivo* at 7TPLOS ONE

Dear Dr. Jacobs,

Thank you for submitting your manuscript to PLOS ONE. After careful consideration, we feel that it has merit but does not fully meet PLOS ONE’s publication criteria as it currently stands. Therefore, we invite you to submit a revised version of the manuscript that addresses the points raised during the review process.

I have decided on a major revision based on the reports from two experienced reviewers. It is an interesting study, but with limited number of subjects and uncertainty in experimental design. Please clarify and respond to the points raised by both reviewers.

We look forward to receiving your revised manuscript.

Kind regards,

Peter Lundberg

Academic Editor

PLOS ONE

Journal Requirements:

"This work was supported by a pilot grant from the Thomas B. and Jeanette E. Laws McCabe Fund to RPRN, the National Institutes of Biomedical Imaging and Bioengineering of the National Institutes of Health under Award Number P41EB029460 to RR, the National Institute of Mental Health of the National Institutes of Health under Award Numbers MH120174 to DR and MH119185 to DR, and the National Institute of Drug Abuse of the National Institutes of Health under Award Number R01 DA037289 to CNE and R01 DA018359 to CNE. "

"I have read the journal's policy and the authors of this manuscript have the following competing interests: Dr. Epperson discloses that she has received research grant support from Sage Therapeutics and HealthRhythms. She is a consultant for Sage Therapeutics, BabyScripts, and Asarina Pharma."

Reviewers' comments:

Reviewer's Responses to Questions

**Comments to the Author**

1. Is the manuscript technically sound, and do the data support the conclusions?

Reviewer #1: Yes

Reviewer #2: Partly

2. Has the statistical analysis been performed appropriately and rigorously? 

Reviewer #1: Yes

Reviewer #2: No

3. Have the authors made all data underlying the findings in their manuscript fully available?

Reviewer #1: Yes

Reviewer #2: No

4. Is the manuscript presented in an intelligible fashion and written in standard English?

Reviewer #1: Yes

Reviewer #2: Yes

5. Review Comments to the Author

Reviewer #1: The goal of the study was to determine the effectiveness of GluCEST as a biomarker for nicotine use, as nicotine will block the glutamate receptors. This was done in a small population of smokers versus non-smokers. Whole brain, grey matter, white matter, and seven cortical regions, were analyzed in one imaging slice. The results were nonsignificant with respect to GluCEST differences between the two groups in these regions. The only indication is a moderate negative correlation with age in one region for smokers, but no statistical significance is mentioned.

The paper is well organized, but the small number of participants and limited requirement for the number of cigarettes smoked probably affected the finding of correlations. However, the limitations of the study are well described. I believe it is important to report negative results, but the below minor comments need to be addressed.

1) The abstract states, “T1-weighted images were used to segment the GluCEST images into seven cortical and four subcortical regions”. However, the methods/results only specify seven subcortical regions. Can you please explain this?

2) What is the statistical significance of the negative age correlation found in one region?

3) The binding of nicotine to the glutamate receptor should increase the glutamate concentration. So why would there be a negative age dependence in smokers?

4) Discussion, 2nd paragraph “were observed” done twice.

5) How about other contributions to the GluCEST signal? Could these be a limitation?

Reviewer #2: Review comments:

Previous studies employing various techniques have yielded conflicting results regarding regional glutamate differences between smokers and non-smokers. In this study, the potential of GluCEST imaging as a biomarker for nicotine addiction was explored. The researchers observed similar GluCEST levels in both smokers and non-smokers, with a moderate negative age correlation specifically in the cingulate gyrus. However, no statistically significant differences were noted between smokers and non-smokers across measured cortical or subcortical regions.

To enhance the clarity of the paper, the authors should address the following key issues:

1. Analysis of Conflicting Literature:

In the context of this research, the broader literature on regional glutamate differences in smokers versus non-smokers presents conflicting findings. The authors should delve into potential factors contributing to these discrepancies and outline their proposed solutions. Notably, previous studies were conducted at 4T field strength and below, utilizing various MRS techniques. While GluCEST imaging offers simultaneous measurement advantages, the authors should acknowledge the limitations and potential confounding factors, such as variability in ROI definitions, total intracranial volume, stability of the measurements and gender difference.

2. Sample Size Justification:

The authors need to elaborate on their rationale for selecting a relatively small sample size. A thorough discussion on how this choice may impact the robustness of their conclusions is essential. Additionally, addressing whether the sample size aligns with statistical power considerations would contribute to the overall validity of the study.

3. Consideration of Additional Biological Differences:

Beyond glutamate levels, the authors should explore other quantitatively measurable biological differences between smokers and non-smokers, such as grey matter volume and nicotine concentrations. Including these factors as covariates in the statistical analysis could provide a more comprehensive understanding of the observed GluCEST imaging results.

4. Detailed Methodological Information:

Provide more comprehensive details on tissue segmentation, ROI definition and size, and the positioning of GluCEST imaging slices. In particular, clarify how the authors standardized the definition of the same oblique axial slice across different subjects. This information is crucial for replicability and understanding the precision of the methodology employed.

5. Proofreading and Terminology:

Conduct a thorough proofread to correct any grammatical errors and spelling mistakes, such as 'brian' instead of 'brain.' Ensuring precise and accurate terminology is used throughout the paper contributes to the overall professionalism and credibility of the research.

6. PLOS authors have the option to publish the peer review history of their article (what does this mean?). If published, this will include your full peer review and any attached files.

Reviewer #1: No

Reviewer #2: **Yes: **Tie-Qiang Li

---

## [Author Response · Author response to Decision Letter 0]

18 Dec 2023

We thank all the reviewers for providing constructive feedback and points of improvement to our manuscript. We believe the suggested changes have strengthened the overall manuscript significantly. We provide a point-by-point response to the reviewer's comments as an attached document.

---

## [Editor Report · Decision Letter 1]

3 Jan 2024

Application of glutamate weighted CEST in brain imaging of nicotine dependent participants *in vivo* at 7T

PONE-D-23-22757R1

Dear Dr. Jacobs,

We’re pleased to inform you that your manuscript has been judged scientifically suitable for publication and will be formally accepted for publication once it meets all outstanding technical requirements.

Kind regards,

Peter Lundberg

Academic Editor

PLOS ONE
---

## [Editor Report · Acceptance letter]

9 Feb 2024

PONE-D-23-22757R1 

PLOS ONE

Dear Dr. Jacobs, 

I'm pleased to inform you that your manuscript has been deemed suitable for publication in PLOS ONE. Congratulations! Your manuscript is now being handed over to our production team.

Kind regards, 

on behalf of

Professor Peter Lundberg 

Academic Editor

PLOS ONE